# Essential Protein PHB2 and Its Regulatory Mechanisms in Cancer

**DOI:** 10.3390/cells12081211

**Published:** 2023-04-21

**Authors:** Amanda Qi, Lillie Lamont, Evelyn Liu, Sarina D. Murray, Xiangbing Meng, Shujie Yang

**Affiliations:** 1Department of Pathology, Carver College of Medicine, University of Iowa, Iowa City, IA 52242, USA; amanda-qi@uiowa.edu (A.Q.); lillie-lamont@uiowa.edu (L.L.); evelynliu500@gmail.com (E.L.); sarina-murray@uiowa.edu (S.D.M.); xiangbing-meng@uiowa.edu (X.M.); 2Holden Comprehensive Cancer Center, Carver College of Medicine, University of Iowa, Iowa City, IA 52242, USA

**Keywords:** prohibitins, PHB2, essential gene, anti-apoptosis, mitophagy

## Abstract

Prohibitins (PHBs) are a highly conserved class of proteins and have an essential role in transcription, epigenetic regulation, nuclear signaling, mitochondrial structural integrity, cell division, and cellular membrane metabolism. Prohibitins form a heterodimeric complex, consisting of two proteins, prohibitin 1 (PHB1) and prohibitin 2 (PHB2). They have been discovered to have crucial roles in regulating cancer and other metabolic diseases, functioning both together and independently. As there have been many previously published reviews on PHB1, this review focuses on the lesser studied prohibitin, PHB2. The role of PHB2 in cancer is controversial. In most human cancers, overexpressed PHB2 enhances tumor progression, while in some cancers, it suppresses tumor progression. In this review, we focus on (1) the history, family, and structure of prohibitins, (2) the essential location-dependent functions of PHB2, (3) dysfunction in cancer, and (4) the promising modulators to target PHB2. At the end, we discuss future directions and the clinical significance of this common essential gene in cancer.

## 1. Introduction

### 1.1. History of Prohibitin

The *prohibitin* (*PHB*) gene was initially found to be antiproliferative and able to inhibit the initiation of DNA synthesis in rat liver in 1989 [1]. The human homologue (later to be known as PHB1) was then identified, cloned, and mapped to the human chromosome 17q21 [2,3]. In 1994, another member of prohibitin, PHB2, was discovered on chromosome 12p13 when two proteins were found to associate with the IgM antigen receptor of B lymphocytes. Otherwise known as B cell receptor-associated proteins (BAPs), amino acid sequencing of BAP32 and BAP37 identified them to be PHB1 and PHB2, respectively [4]. Another name for PHB2 denoted as REA (repressor of estrogen receptor activity) was found to inhibit estrogen receptors’ (ERs) activity [5]. DNA sequencing showed that REA had the same identity as BAP37 [5]. The history of PHB2 was summarized in Figure 1.

### 1.2. Prohibitin Family and Structure

Prohibitin (PHB) 1 and 2 belong to the Stomatin/prohibitin/flotillin/HflKC (SPFH) family, which also includes Erlin-1 and 2, podocin, stomatin, and flotillin-1 and 2. The main function of this family contributes to the formation of membrane microdomains and lipid raft-associated processes [6]. The human *PHB* genes encode two protein isoforms, PHB1 and PHB2, with molecular weights of 32 and 34 kDa, respectively. They both contain an evolutionarily conserved prohibitin N-terminal transmembrane domain involved in scaffolding and a C-terminal coiled-coil domain for protein–protein interactions [7]. PHB2 contains three different domains including a transmembrane domain required for mitochondrial localization, a central prohibitin domain, and an overlapping coiled-coil domain as shown in Figure 2 [3].

They are both expressed in the mitochondria, nucleus, and cell membrane, and are involved in the processes of cell proliferation, apoptosis, mitophagy, and metastasis. In the inner mitochondrial membrane (IMM), PHB1 and PHB2 form heterodimers assembled in ring-shaped complexes to maintain mitochondrial stability but are exported out upon signaling [8]. In the nucleus, PHB1 and PHB2 work independently to modulate transcriptional activity [9].

### 1.3. Prohibitin Expression

High expressions of *PHB1* and *PHB2* were reported in 1078 cell lines in the Cancer Cell Line Encyclopedia (CCLE) and were recognized as common essential genes in 1077 of the 1078 tested cell lines (Depmap.org/Portal). A recent publication also confirmed that *PHB1* and *PHB2* are common essential genes with potential mechanisms involved in protein translation initiation and transfer RNA (tRNA) ligases [10]. The average *PHB2* expression is high in 1078 tested cell lines and *PHB2* high expression can be derived from amplification. 

In a 2018 review paper, expression of PHB1 and PHB2 was surveyed in 17 cancer types. It was reported that PHB1 and PHB2 expression levels are high in most cancer types, for both mRNA and protein [11].

### 1.4. PHB1

Initial studies found that PHB1 induced cell cycle arrest and inhibited cell proliferation. However, recent studies of PHB1 in esophageal squamous cell carcinoma (ESCC), gallbladder cancer, and bladder cancer exhibit promotion of cancer cell proliferation. The role of PHB1 in cancer remains inconsistent. PHB1 has been well studied and summarized in multiple review papers [11,12,13] and will be less focused on in this review paper. Instead, this review will focus on PHB2 and its role in cancer. 

## 2. Critical Functions of PHB2

### 2.1. Conditional Depletion of PHB2

PHB2 is essential for cell survival. Homozygous knockout (ko) PHB2 is lethal, which demonstrates that PHB2 is indispensable for development [14,15]. Therefore, conditional ko PHB2 has been investigated in multiple model systems and has been proven to be crucial for maintaining organ function.

In pancreatic β-cells, ko PHB2 promotes mitochondrial damage and impairs insulin secretion through loss of β-cells. Thus, ko PHB2 decreases survival and induces gradual diabetes [14]. As for the heart, ko PHB2 hinders cardiac fatty acid oxidation, leading to heart failure. Conditional ko *PHB2* mice had severe dilated cardiomyopathy and heart failure due to lipid droplet accumulation and mitochondrial dysfunction. Thus, PHB2 is thought to downregulate carnitine palmitoyltransferase1b (CPT1b), which is the most important enzyme of cardiac fatty acid oxidation [15]. Furthermore, forebrain specific ko PHB2 mice exhibited neurodegeneration, leading to behavioral and cognitive deficiencies. PHB2 demonstrated critical functions for the survival of neurons through mitochondrial fusion and ultrastructure [16]. Lastly, hepatocyte-specific ko PHB2 mice exhibited lipid accumulation, impaired gluconeogenesis, and liver failure. Similarly, liver PHB2-deficiency led to mitochondrial liver dysfunction [17]. Therefore, PHB2 is critical for organ development in the pancreas, heart, forebrain, and liver, because loss of PHB2 leads to organ deterioration. 

### 2.2. Location-Dependent PHB2 Function

PHB2 is often located on the cell membrane, in the mitochondria, and in the nucleus, with distinct roles in different locations. These location-dependent functions will be described in this section and in Figure 3. 

#### 2.2.1. PHB2 Function on Cell Membrane

PHB2 was found to be present and expressed in multiple cellular compartments, including the plasma membrane, and was also found to be associated with specific cell membrane receptors [9,18]. PHB2 originates from an evolutionarily conserved family of proteins, including stomatins, flotillins, and the human insulin receptor [18]. Furthermore, stomatin-like proteins regulate ion channels and mechanosensation; flotillins affect signaling across the plasma membrane and regulate membrane curvature; erlins target inositol (1,4,5)-trisphosphate receptors in the ER membrane for degradation; and prohibitins and bacterial HflK/C proteins associate with AAA proteases and are involved in proteolytic processes [19]. The role of PHB2 in signal transduction at the plasma membrane was found by the association of PHB2 with the IgM receptor in B cells [19].

Another protein found to be associated with PHB2 is Serine Incorporator 3 (SERINC3), a protein primarily localized at the plasma membrane. SERINC3 was identified in complex with PHB2 by coimmunoprecipitation and Western blotting [9]. Additional cytoplasmic proteins, such as insulin-like growth factor 1 (IGFR1), and integrins, such as vascular cell adhesion protein 1 (VCAM1), were found to interact with PHB2 and are associated with integral cell membrane proteins and cellular receptors [9]. Interestingly, the identification of PHB2 and the plasma membrane were made in infectious disease studies. The PHB1:PHB2 complex was found to associate with amino acid residues 790 to 800 in the carboxyl terminus of the human immunodeficiency virus (HIV) glycoprotein [9]. These findings suggest that PHB2 can regulate nucleolus function directly as well as via transduction pathways, which may be related to the role of PHB2 in the regulation of cell proliferation [20]. 

#### 2.2.2. PHB2 Function in Mitochondria 

PHB2 plays a critical role in metabolically active cells, as suggested by the high expression levels in neurons, muscle, heart, and liver cells, all of which massively rely on mitochondrial function [18]. Not surprisingly, prohibitins have been discovered to be associated with countless mitochondrial mechanisms, including mitochondrial respiratory chain subunit degradation, assembly, activity of the oxidative phosphorylation system (OXPHOS), mitochondrial biogenesis, mitochondrial apoptosis, and mitophagy [7].

Mitophagy is a selective autophagy process specifically involved in the degradation of damaged or redundant mitochondria in cells to maintain cellular homeostasis [21]. Mitophagy may promote survival through the adaptation to stress by removing mitochondria that could be permeabilized to induce cell death or, conversely, it may result in cell death from the excessive elimination of mitochondria. This degradation of mitochondria has contrasting implications on tumor growth and development, making the role of mitophagy, regarding cellular fate, variable [22].

A 2017 paper published in Cell discovered that PHB2 is a IMM mitophagy receptor. It is involved in parkin-induced mitophagy by binding the autophagosomal membrane-associated protein LC3 through PHB2’s LC3-interacting region (LIR) domains. In vivo, PHB2 has an essential role in eliminating the transmission of paternally derived mitochondrial DNA to offspring in *C. elegans*. The mechanism through which PHB2 regulates mitophagy in paternal mitochondrial DNA is unknown. However, this confirms PHB2’s multiple roles in mitophagy, acting as a mediator of mitochondrial function as well as IMM binding to LC3 [23,24]. 

Yan et al. reported that in addition to functioning as a mitophagy receptor, PHB2 enhances PTEN-induced kinase 1 parkin (PINK1-PRKN)-induced mitophagy. To initiate mitophagy, PHB2 binds to PARL (presenilin-associated rhomboid-like protein), releasing PGAM5 (phosphoglycerate mutase 5) in the process, which is responsible for stabilizing PINK1 (PTEN induced putative kinase 1) on the outer mitochondrial membrane (OMM) [25]. 

Prohibitins also play a crucial role in inhibiting apoptosis, the process of programmed cell death. Apoptosis is essential for cells to maintain regular physiological activity [12]. Dysfunction of the mitochondria-regulated apoptosis pathway leads to cancer proliferation [26]. PHB2 interacts with optic atrophy 1 protein (OPA1), an IMM protein, which is responsible for mitochondrial fusion and cristae integrity. OPA1 influences resistance to apoptosis by regulating mitochondrial cristae remodeling [27]. Dysregulation of OPA1 processing leads to apoptosis resistance. OPA1 has two forms: long (L-OPA1) and short (S-OPA1), through proteolytic processing. Loss of L-OPA1 and accumulation of S-OPA1 results in mitochondrial fragmentation and apoptosis. OPA1 proteolytic processing is protected by PHB2 in a chaperone-like manner. It was observed that depletion of PHB2 accelerated the cleavage of L-OPA1 into S-OPA1 [17,28]. Thus, PHB2 increases resistance to apoptosis, exerting its anti-apoptotic function via stabilizing OPA1 [28].

PHB2 was also found to directly bind with Hax-1 (HCLS1-associated protein X-1), an anti-apoptotic protein, in the mitochondria. Similarly to OPA1, depletion of PHB2 leads to an increase in the degradation of Hax-1, inducing caspase-dependent apoptosis. Reduced mitochondrial Hax-1 is associated with an increased loss of mitochondrial integrity, through the activation of the caspase 9/caspase 3 pathway, resulting in apoptosis. When Hax-1 was knocked down, apoptosis occurred without causing damage to the morphology of the mitochondria. With this, it can be concluded that modified levels of PHB2 cause a decrease in mitochondrial fusion, and an increase in the formation of reactive oxygen species (ROS) and cell death [7]. 

Recent discoveries have brought to light the various roles that prohibitins have in cancer, many of which are linked to the elevated metabolic reliance on mitochondria respiration. Generally, cancer cells have a higher overall energy demand that is reached through an increase in cellular respiration and glycolysis. This increase leads to a cellular disproportion of ROS and Ca^2+^ stores. Cancer cells require maximum mitochondrial function, and in turn, their mitochondria are especially vulnerable to oxidative damage. In these circumstances, an increase in the expression of prohibitins typically results in an increase in the stability of mitochondria. The results are not always consistent. In one study, knockdown of prohibitins led to a decrease in cell division and no change in mitochondrial integrity, while another study showed that knockdown of PHB2 led to a decrease in cell proliferation as well as a major decrease in mitochondrial integrity [9].

PHB2 also forms a tripartite complex with AURKA (aurora kinase A) and LC3 (gene name is *MAP1LC3*, microtubule-associated proteins 1 light chain 3). The complex induces mitophagy, following phosphorylation of PHB2 on Ser39. This is referred to as AURKA-dependent mitophagy and is not correlated with the PARK2–Parkin/PINK1 pathway. Xanthohumol, a PHB2 ligand, blocks the formation of this complex by changing the distance between AURKA and LC3. Due to this, the binding of xanthohumol prevents AURKA-dependent mitochondrial loss and prevents the excessive generation of ATP that is associated with high levels of AURKA expression. This illustrates that a metabolic change occurs when AURKA is overexpressed and demonstrates the connection between mitophagy and cancer cell’s metabolic capacity [29]. 

Apart from OPA1, Hax-1, and AURKA, there are many other proteins that bind PHB2 in the mitochondria, including m-AAA, OMA1 proteases, and SLP-2. PHB2 also regulates numerous mitochondria functions not mentioned, such as oxidative phosphorylation, mitochondrial biogenesis, and unfolded protein response. These have been summarized and referenced in a 2019 review paper [7]. 

#### 2.2.3. PHB2 Function in Nucleus 

PHB1 has critical functions in the nucleus in three areas: (1) repressing transcription, (2) inducing p53 mediated transcription, and (3) mediating RAS signal transduction (through its interaction with Raf1) [30]. In comparison with PHB1, PHB2 has similar functions in the nucleus as a mediator of transcription with various transcription factors and nuclear receptors. 

Within the nucleus, PHB2 plays an essential role in the regulation of transcription factors, including estrogen receptor, myocyte enhancer factor 2 (MEF2), myoblast determination protein 1 (MyoD), and peroxisome proliferator-activated receptor (PPARγ) [31,32,33]. PHB2 was discovered to inhibit COUP-TF1 and II, orphan nuclear hormone receptors (NHRs), through recruitment of histone deacetylases [34]. As COUP-TF1 and II are both pertinent to the development of embryos and the immune system, aberrant expression of PHB2 would have catastrophic repercussions regarding development [34].

Prohibitins, PHB1 and PHB2, are also significantly linked to histone deacetylases (HDACs). To repress transcription of nuclear receptors, PHB2 associates with HDAC1 and HDAC5 [34]. PHB2 recruits HDACs to the nucleus, further inhibiting the transcription of nuclear receptors. Due to this, HDAC inhibitors can prevent transcriptional inhibition, which is typically regulated by PHB2. Therefore, PHB2 can inhibit transcription through both its immediate interaction with nuclear receptors and through the recruitment of HDACs to the transcription site. 

PHB2 was also found to interact with two common muscle regulation factors, MyoD and MEF2 [33]. PHB2 represses their transcriptional activity by the formation of a complex with both MyoD and MEF2 in the nucleus. Yeast-2 hybrid experiments illustrated PHB2’s direct interaction with MyoD and its indirect interaction with MEF2. These interactions were shown to be repressed by PHB2’s association with AKT2 (AKT serine/threonine kinase 2) [9,33]. Although this study illustrated that AKT2 interacts with PHB2, no modification of PHB2 by AKT2 was observed during myocytic differentiation [33]. Therefore, it was concluded that the disruption of the PHB2: MyoD: MEF2 interaction is not correlated to AKT2’s catalytic function [33]. 

A cotranscriptional activator of PPARγ, PGC-1α, has also been shown to interact with PHB2. This interaction inhibits the transcriptional activity of PPARγ [31]. PPARγ functions as a transcriptional regulator of many genes involved in activating adipogenesis. PHB2 levels are increased during adipogenesis; if PHB2 is knocked down, inhibition of adipogenesis is observed [9]. 

## 3. PHB2 Tumor Suppressor Function 

PHB2 was discovered as a tumor suppressor in breast cancer, osteosarcoma, and head and neck squamous cancer. The main mechanisms include inhibition of ERα, interaction with BIG3 (brefeldin A-inhibited guanine nucleotide-exchange protein 3), and promotion of *PIG3* (p53-inducible gene 3) transcription. 

### 3.1. Breast Cancer (BC)

PHB2 executes tumor suppressive activity in breast cancer (BC) through its involvement with ERα (estrogen receptor-alpha). Estrogen is a well-known proliferation factor in BC. Its binding to ERα significantly increases the proliferation and metastasis of BC tumor cells. PHB2 has been found to translocate to the nucleus in the presence of estrogen and ERα. In the nucleus, it mediates transcription, functioning as a corepressor of ERα [34].

BIG3 is an E2/ERα signal regulator that is overexpressed in breast cancer. *BIG3* is regulated by ERα in a positive feedback loop. It interacts with PHB2 in the cytoplasm, inhibiting its nuclear translocation. The potential mechanism of BIG3 binding PHB2 is through its specific loop region [35]. This activates the ERα-signal transduction pathways, promoting cancer proliferation [36]. 

In 2017, the Katagiri group reported that ERAP (ERα activity-regulator synthetic peptide derived from α-helical BIG3 sequence, 165–177 amino acids) is a cell-permeable peptide inhibitor shown to inhibit BIG3-PHB2 binding. ERAP also downregulates *BIG3* transcription to reduce BIG3 levels in the cytoplasm of cancer cells. ERAP competitively binds BIG3 and then promotes PHB2 translocation into the nucleus to facilitate the direct binding of ERα [36]. A chemically modified ERAP, stapled ERAP (stERAP) has a longer duration to inhibit breast tumorigenesis. An in vivo study illustrated that stERAP increases the suppression of E2-induced tumor proliferation in mice via prolonged inhibition of BIG3-PHB2 interaction because of a higher PHB2-binding affinity compared with that of unstapled ERAP [37]. This study suggests that targeting the BIG3-PHB2 interaction is a promising treatment strategy to inhibit luminal-type BC growth. The interaction of PHB2, BIG3, and ERAP was summarized in Figure 4.

Progesterone receptor membrane component 1 (PGRMC1) has implications with ERα activation. Prohibitins interact with PGRMC1 with dependence on S181-phosphorylation upon treatment with proliferation-promoting progestins [38]. Phosphorylated PGRMC1 associates with PHBs, which prevents PHBs binding to ERα. In summary, PHBs play an important role in inhibiting ERα and breast cancer tumorigenesis, however, PGRMC1 hinders their antiproliferative function [38].

In a 2022 paper, it was discovered that PHB2 in the cell membrane may function as a biomarker for precise diagnosis of the luminal A breast cancer cell subtype. This marker would allow for the differentiation of the luminal A breast cancer cell subtype from HER2-positive and triple-negative breast cancer cell subtypes [39]. 

Another paper reported a controversial oncogenic function of PHB2 in MCF7 (a breast cancer cell line with estrogen, progesterone, and glucocorticoid receptors) cells [40]. Fluorizolines, a PHB1/2 specific modulator [41,42], interrupt the interaction between PHB2 and γ-glutamylcyclotransferases (GGCT) and enhance p21 expression. Fluorizoline treatment prevents PHB2’s translocation into the nucleus and inhibits cell cycle progression. The authors did not acknowledge PHB2’s tumor suppressor function in breast cancer as reported previously, and no explanation of the discrepancy was discussed in this paper.

### 3.2. Osteosarcoma (OS)

In 2021, the Katagiri group extended the study of the BIG3-PHB2 interaction in osteosarcoma (OS). Much like breast cancer, overexpression of BIG3 was also discovered in OS [43]. Unlike breast cancer cells, however, the BIG3-PHB2 complex was found to be mainly localized in the mitochondria of OS cells. Inhibition of the formation of the BIG3-PHB2 complex using siRNA was shown to repress the proliferation and migration of OS cells. The knockdown of BIG3 resulted in a substantial decrease in OS cell proliferation. This study also illustrated that the BIG3-PHB2 complex may regulate apoptosis, which depends on the PARP1/AIF pathway via direct binding to PARP-1 (Poly [ADP-ribose] polymerase 1) in OS cells. Consequently, disruption of this complex would promote apoptosis of OS cells [43]. More importantly, it was approved that stERAP treatment significantly decreases OS tumor growth by inhibition of BIG3-PHB2 complex formation and induced apoptosis in vivo. 

### 3.3. Head and Neck Squamous Cancer Cells

Prohibitins, PHB1 and PHB2, function as tumor suppressors in head and neck squamous cell carcinomas. The presence of prohibitins is positively correlated with PIG3-mediated apoptosis. PHB1 and PHB2 bind to the *PIG3* (TGYCC)_15_ promoter. In the presence of apoptotic stress, PHB1 and PHB2 recruit p53 to the *PIG3* gene, leading to p53-induced *PIG3* transcription and PIG3-mediated apoptosis. Even in the absence of apoptotic stress, prohibitins also bind the *PIG3* promoter, initiating *PIG3* transcription and apoptosis in the absence of p53 [44]. 

## 4. PHB2 Oncogenic Function

PHB2 was reported as an oncogene by a variety of mechanisms in 12 cancer types. PHB2 has four main functions: (1) promote migration, (2) enhance proliferation, (3) induce anti-apoptosis and cellular survival, and (4) initiate mitophagy. The detailed mechanisms in different cancers were summarized in Table 1 and illustrated in Figure 5.

### 4.1. Prostate Cancer (PCa)

PHB2 is known to promote prostate cancer (PCa) progression and is overexpressed in PCa. Data from a wound healing assay supported PHB2 being a positive regulatory factor for PCa cell migration [45]. AKT2, a promigratory kinase, was found to be correlated with the expression of PHB2. AKT2 is well known for its proliferative role in breast cancer but has been shown to be a negative regulator of PCa cell migration in PC3 cells [64]. It was found that overexpression of PHB2 reduced AKT2 expression levels and knockdown of PHB2 increased the expression of AKT2 levels in prostate cancer cells [45]. Moreover, the data illustrated that co-overexpression of PHB2 and AKT2 caused a substantial decrease in PCa cell migration. This data is consistent with data reported in muscle cells, where feedback regulation between AKT2 and PHB2 upheld a dynamic balance. Although PHB2’s overall role in PCa proliferation is not well studied, the collected data supports the proposed mechanism of its inhibitory effect on AKT2 expression [45,65]. 

### 4.2. Non-Small Cell Lung Cancer (NSCLC)

PHB2 has been found to be overexpressed in NSCLC tumors. PHB2 overexpression promoted proliferation, migration, and invasion, whereas PHB2 knockdown enhanced apoptosis. PHB2 promotes tumorigenesis in NSCLC by interacting with and stabilizing receptors for activated C kinase 1 (RACK1). RACK1 plays an important role in cancer progression and tumorigenesis. RACK1 is a highly conserved intracellular adaptor protein and an important mediator of various cellular pathways. PHB2 increases RACK1 expression through posttranslational modification and induces the activation of downstream tumor-promoting effectors. The increase in RACK1 expression increases activation of the AKT and FAK pathways through integrin β1 [46].

When under metabolic pressure or hypoxia, mitophagy, the breakdown of dysfunctional mitochondria, may provide protective effects to malignant tumors. When mitochondrial dysfunction occurs, mitophagy reduces the amount of ROS in the cell, limiting cell death, and the removal of mitochondria provides more nutrients for tumor cells. PHB2 has been seen to contribute to parkin-mediated mitophagy in NSCLC cell lines. Under mitochondrial stress, PINK1 phosphorylates parkin, which ubiquitinates OMM proteins, leading to the breakdown of OMM. This exposes the IMM and IMM proteins, such as PHB2. PHB2 binds to the autophagosome LC3 II protein and induces mitophagy, though the exact pathway from complex formation to mitophagy is unclear [47]. Thus, PHB2 is a significant mitophagy receptor in the mitochondria and a positive modulator of RACK1 in the cytoplasm. 

### 4.3. Colon Cancer (CRC)

In colorectal tumor cells, there is an overexpression of both PHB2 and thymidylate synthase (TS). TS is an S-phase enzyme responsible for catalyzing the methylation of dUMP to produce dTMP, a precursor of DNA synthesis. TS functions as an oncogene; its expression was found to be correlated with colorectal cancer prognosis [48]. Rabdosianone I, a drug extracted from an oriental herb, has been seen to reduce TS expression in colorectal cancer patients and to bind both PHB2 and ANT2 [48]. ANT2 and PHB2 were shown to stabilize TS at the protein level with PHB2 also possibly promoting *TS* transcription with increased *TS* mRNA levels. Rabdosianone I bound to both ANT2 and PHB2 caused proteasomal degradation of the TS protein with *TS* transcription being inhibited when bound to PHB2 alone [48]. 

Overexpression of PHB2 enhances colon cancer growth in vitro and in vivo. PHB2 was reported as a novel target of Dihydroartemisinin (DHA) in colon cancer [49]. DHA is an effective clinical drug for treatment of malaria, but it also has promising anticancer effects that have been tested in many cancers [66]. The proposed mechanism is through DHA inhibiting the expression of PHB2 in a ubiquitylation-dependent manner and reversing PHB2-induced RCHY1 (RING finger and CHY zinc finger domain-containing protein 1) expression, as well as suppressing p53 and p21 expression. Therefore, both PHB2 and RCHY1 are promising targets in colon cancer therapy [49]. 

PHB2 promoting tumorigenesis in CRC was also confirmed by another group. Ren et al. reported that PHB2 stabilizes mitochondria complex 1 and increases its activity by binding a mitochondria protein, NDUFS1 (NADH: ubiquinone oxidoreductase core subunit S1). This binding boosts OXPHOS levels and enhances cell proliferation and tumorigenesis. This finding proposes that PHB2 is a key player for CRC energy metabolism and a potential target for CRC therapy [50].

Recent studies have placed an emphasis on the importance of long non-coding RNAs (lncRNAs) in regulating cancer and stem cells. One study highlighted the important interaction between PHB2 and a specific lncRNA tied to colon cancer, Lnc34a [51]. Lnc34a is enriched in colon cancer stem cells and initiates tumorigenesis by targeting miR-34a, a well studied miRNA tumor suppressor that has a synergistic effect with p53 [67]. This suppressed activity is associated with slower growth of varying tumors, such as those of lung, bowel, and CRC. In CRC-advanced tumors, miR-34a is typically missing due to methylation on its promoter, leading to its downregulation [51]. 

The potential mechanism of miR-34a methylation was investigated by performing a RIP (RNA Immunoprecipitation). The RIP data revealed that Lnc34a interacts with both PHB2 and HDAC1, while recruiting DNMT3A via PHB2 [51]. When PHB2 was knocked down, DNMT3A (but not HDAC1) no longer suppressed miR-34a expression. A mechanistic study revealed that Lnc34a binds to the miR-34a promoter, recruiting PHB2/DNMT3A and HDAC1 to methylate and deacetylate the promoter, leading to miR-34a silencing and CRC proliferation [51].

### 4.4. Hepatocellular (HCC)

In hepatocellular carcinoma (HCC), PHB2 promotes cell proliferation by aiding in the repression of miR-34a production. Low levels of miR-34a are associated with higher occurrences of bone metastasis (BM) and poor patient prognosis [52]. As discussed in colon cancer (Section 4.3), Lnc34a binds to PHB2 and forms a complex with DNMT3A, methylating the miR-34a promoter. Lnc34a then induces HDAC1 to promote histone deacetylation, which inhibits miR-34a transcription. The overexpression of Lnc34a in HCC cells is associated with an increase in BM through repressing miR-34a, which typically has an inhibitory effect on HCC proliferation and BM [52]. 

A publication on hepatocellular cancer by the Xiang group reported PHB2 as a key player in PINT87aa-induced cancer senescence. PINT87aa is peptide encoded by the p53 induced transcript (*LINC-PINT*). Overexpression of PINT87aa in HCC cells was shown to induce cell senescence by binding to FOXM1 (Forkhead box protein M1). This binding blocks *PHB2* transcription, preventing mitophagy [53]. Overall, both the antiproliferation and pro-senescence activities of PINT87aa in HCC cells were reported, with PHB2 being a key mediator in these functions. This provides rationale for PINT87aa as a potential anti-cancer drug for HCC [53].

### 4.5. Leukemia

In human myeloid leukemia cells, it has been shown that capsaicin directly binds to PHB2, inducing PHB2 nuclear translocation and dissociation of the PHB2-ANT2 complex. The depletion of PHB2 in the mitochondria compromises mitochondrial stability, causing the release of cytochrome c, the inhibition of ADP uptake by ANT2, and the induction of apoptotic cell death [54]. 

Additionally, fluorizoline, an antiproliferative drug, has been tested and seen to be effective in acute myeloid leukemia cells. It is hypothesized that fluorizoline increases NOXA levels in the presence of PHB2, inducing cell death through the p53-mediated apoptotic pathway [55]. 

Increased AKT activity was often reported in many cancers as AKT enhances the metabolic process and promotes cell survival. In leukemia, PHB2 was found to interact with AKT. Both AKT1 and -2 were discovered to phosphorylate PHB2 on Ser-91 [56]. The forced PHB2 (S91A) mutant resulted in rapid apoptotic cancer cell death. Phosphorylated PHB2 may regulate nuclear–mitochondrial activity, playing a crucial role in cell survival during differentiation. Both in vivo and in vitro, PHB2 was complexed with AKT in the nucleus of leukemia cells and phosphorylated during differentiation induced by all- trans retinoic acid (ATRA) treatment [56]. 

A recent paper illustrated that, consistent with the proliferative function of PHB2 in leukemia cell lines, overexpression of PHB2 was correlated with adverse prognosis in cytogenetically normal acute myeloid leukemia (CN-AML patients). Due to this, PHB2 may function as an independent prognostic marker for CN-AM, although the mechanism was not fully studied in this report [68]. 

### 4.6. Ovarian Cancer

Interactions between the mitochondrial inner and outer membranes have significant impacts on the sensitivity of ovarian cancer to cisplatin. Cisplatin is a strong chemotherapy drug commonly used to treat ovarian cancer. Activation of OMA1, a mitochondrial protease, was shown to increase ovarian cancer sensitivity to cisplatin, both in vitro and in vivo [57]. PHB2 was discovered to form a complex with STOML2 (stomatin-like protein 2) in the IMM to maintain its stability. The STOML2/PHB2 complex regulates mitochondrial protease activity, including OMA1. As Cheng et al. reported in 2022, destruction of the PHB2/STOML2 complex promotes the protease activity of OMA1 [57]. PHB2 and STOML2 are both reportedly overexpressed and attributed to the mitochondrial function and apoptosis resistance in ovarian cancer.

OMA1 cleaves both OPA1 and DELE1 (DAP3 Binding Cell Death Enhancer 1). OPA1 cleavage results in the remodeling of the mitochondrial cristae and cleavage of DELE1 induces its cytoplasmic movement. In the cytoplasm, the cleaved DELE1 interacts with EIF2AK1 (eukaryotic translation initiation factor 2 alpha kinase 1) to promote the transcription of pro-apoptotic proteins [57]. Additionally, the anti-apoptotic protein MCL1 (induced myeloid leukemia cell differentiation protein) is downregulated, resulting in an accumulation of BAX and BAK1 on the OMM. This triggers the degradation of OMM, the release of cytochrome C, and ultimately, an increase in apoptosis [57]. In summary, the breakdown of the PHB2/STOML2 complex promotes apoptosis of ovarian cancer cells through the mitochondrial PHB2/OMA1/DELE1 pathway [57].

### 4.7. Rhabdomyosarcoma (RMS)

PHB2 was found to be localized in the nucleolus and to be positively correlated with RMS proliferation and negatively correlated with differentiation [20]. This finding suggests that PHB2 promotes RMS tumorigenesis. Silencing of PHB2 will induce (1) suppression of cell proliferations evidenced by G1 cell cycle arrest and apoptosis in a small percentage of RD cells, and (2) RMS cell differentiation evidenced by an increase in myogenin-positive RD cells [20]. This supports the idea that PHB2 functions as a repressor for muscle differentiation through inhibiting the transcriptional activities of MyoD and MEF2. Additionally, PHB2 controls the transcription of ribosomal RNA (rDNA) by regulating the access of c-Myc and MyoD to the rDNA promoter. Upon PHB2 knockdown, there was a decrease in the levels of rRNA, suggesting PHB2’s role in regulating nucleus function [20]. Decreased levels of the oncogene Myc were observed at the rDNA promoter when PHB2 was silenced, indicating that PHB2 is necessary for Myc to regulate rRNA’s transcription in RMS cells [20]. In summary, PHB2 localized in the nucleolar of RMS cells regulates rDNA transcription. The PHB2-dependent mechanism increases proliferation by promoting c-Myc activation of rDNA transcription and decreases myogenic differentiation by repressing MyoD binding to rDNA [20].

PHB2 was also shown to bind IGFBP-6 (insulin like growth factor binding protein 6) on the cell surface, regulating the promigratory effect that IGFBP-6 has on RMS cells [58]. Traditionally, IGFBP-6 was reported to have anti-tumorigenic properties through inhibiting the actions of IGF-II (insulin like growth factor 2) and inhibiting angiogenesis via a mechanism independent from IGF. However, IGFBP-6 was reported to have tumorigenic function in RMS [58]. The PHB2-IGFBP-6 binding on the cell membrane causes increased tyrosine phosphorylation of PHB2 at tyrosine 128 and 271 by insulin receptor kinase, which was found to be correlated with cancer cell migration [13]. Rh30 RMS cell migration was completely lost when PHB2 was knocked down [58]. Therefore, targeting PHB2 may combat this adverse effect of IGFBP-6-based treatment for IGF-II-dependent tumors [58]. 

### 4.8. Multiple Myeloma (MM)

*MCC* (mutated in colorectal cancers), a WNT signaling pathway regulator, is a novel oncogene found in B lymphocytes with interactions centering around PARP1 and PHB2. With PHB2 as a hub of the *MCC* interaction network, *MCC* promotes cellular survival and tumorigenesis in multiple myeloma patients [59]. However, the detailed mechanism was not addressed in this paper.

### 4.9. Esophageal Squamous Cell Carcinoma (ESCC)

PHB2 was identified as a potential biomarker for early recurrence/metastasis of ESCC after radical resection. PHB2 was shown to promote cell proliferation and invasion through the AKT signaling pathway. Knockdown of PHB2 downregulated phosphorylation levels of MMP9 (matrix metallopeptidase 9), RAC1, and AKT. Additionally, PHB2 had high expression levels in ESCC tumor tissues, which correlated with an overall worse survival [60]. 

### 4.10. Hematologic: Lymphoid and Myeloid Tumor

Both prohibitins, PHB1 and PHB2, have been found to be associated with T- and B-cell malignancies [61]. PHB1 and PHB2 proteins are overexpressed in lymphoid and myeloid tumor cell lines compared with healthy donor PBMCs (peripheral blood mononuclear cells) [61]. Both PHB1 and PHB2 were found to be primarily occupying the mitochondria in Kit225 cells (a T-cell chronic lymphocytic leukemia patient derived cell line), with the role of maintaining mitochondrial integrity and preventing ROS-mediated cell death [61]. Thus, prohibitins could likely act as biomarkers and molecular targets for treatment of lymphoid and myeloid malignancies [61].

### 4.11. Melanoma

PHB2 regulates LC3/ERK/MITF melanogenic signaling. During melanogenesis, cytosolic LC3 (LC3-I) temporarily binds to phosphatidylethanolamine to form LC3-II [62]. LC3-II activates ERK, leading to the phosphorylation and upregulation of MITF (microphthalmia-associated transcription factor). MITF is a transcription factor that regulates melanocyte development, survival, and function. PHB2 also binds with LC3-II to induce mitophagy as previously discussed in Section 2.2.2 and Section 4.2. Melanogenin analogs, Mel9 and Mel41, bind to PHB2 and activate the conversion of LC3-I to LC3-II. Mel9 and Mel41 also inhibited AKT phosphorylation and downregulated PHBs to promote apoptosis in melanoma cells [62]. 

### 4.12. Pancreatic Cancer

A recent discovery disclosed the tumor proliferative function of PHB2 through its interaction with the oncogene Hes1 [63]. The transcription factor, Hes1, is a downstream effector of Notch signaling. JI051, a small organic molecule, was recently discovered to impair Hes1’s ability of repressing transcription [63]. JI051 treatment reduced the cell growth of the human pancreatic cancer cell line (MIA PaCa-2) [63]. JI051 stabilizes PHB2-Hes1 interaction outside the nucleus, leading to G_2_/M cell-cycle arrest via inhibition of Hes1. JI051 may interfere with Hes1’s function by preventing its nuclear translocation via its interaction with PHB2. It was also observed that ko PHB2 caused a significant decrease in cell proliferation compared with cells with control siRNAs [63]. Data also showed that both PHB2 and Hes1 are necessary for JI051’s effect on cell proliferation, as a suppressed response was observed in the knockdowns of both PHB2 and Hes1. Since PHB2 is overexpressed in pancreatic cancer, it is a potential new target for regulating Hes1 in future cancer treatments [63].

## 5. Reported PHB2 Modulators

As PHB2 promotes tumor progression and contributes to therapeutic resistance in most cancer types, it is a promising therapeutic target. At present, there are 12 small molecular modulators reported to inhibit PHB2’s function. The drug effect and mechanism has been fully described in a 2020 review paper [69] and were summarized in Table 2.

## 6. Discussion

PHB1 and PHB2 have been studied for over 30 years and have proved to be critical proteins with varying functions in different systems. Many questions regarding mechanisms are still unanswered.

### 6.1. Dual Function of PHB2 in Different Cancers

As we discussed in Section 3 and Section 4, PHB2 demonstrates both a tumor repressing function and an oncogenic function. There are three cancer types in which PHB2 functions as a tumor suppressor. In the other 12 cancer types, PHB2 displays as an oncogene (Figure 5). There are no known mechanisms to explain the discrepancy behind this opposite function. Further studies are needed to solve this puzzle.

### 6.2. Shared Molecular Mechanism in Different Cancers

#### 6.2.1. BIG3-PHB2

The interaction between BIG3 and PHB2 (BIG3-PHB2) is important in breast cancer and osteosarcoma. BIG3 binds PHB2 through its loop region and leads to cell proliferation. A peptide inhibitor, ERAP, specifically inhibits BIG3, releasing PHB2 to be translocated into the nucleus (Figure 4). A modified ERAP, stERAP can extend the duration of inhibition, dramatically decreasing tumor growth in both breast cancer and osteosarcoma, as reported by the Katagiri group.

#### 6.2.2. Inhibit miR-34a

MiR-34a is a principal regulator of tumor suppression and is frequently downregulated in multiple cancers [83]. Multiple feedback regulation mechanisms were reported to regulate miR-34a expression. In both colon and hepatocellular cancer, a novel mechanism of miR-34a inhibition was reported where Lnc34a recruits PHB2 to repress miR-34a. Further studies in different cancers are needed to generalize this observation.

#### 6.2.3. Crosstalk with AKT

AKT is a well-studied signaling molecule with three isoforms, AKT1, AKT2, and AKT3, demonstrating distinct functions, either promoting or inhibiting tumorigenesis [84]. In prostate cancer, AKT2 exhibits as a negative regulator of migration function. PHB2 inhibits AKT2 activity and reduces cell migration. However, in NSCLC, AKT promotes cell migration and PHB2 enhances this process. Apart from AKT as a target of PHB2 in prostate cancer and NSCLC, it was also reported as an upstream regulator of PHB2. In leukemia, AKT1 and AKT2 phosphorylate PHB2 at S91, regulate nuclear-mitochondria activity, and further promote cell survival.

### 6.3. PHB2 Location-Dependent Functions (Figure 3)

PHB2 function is location dependent. Outside of the nucleus, PHB2 functions as a signaling protein and scaffolding protein. In mitochondria, PHB1 and PHB2 form ring complexes to regulate cristae, mitophagy, apoptosis, and cell proliferation. Inside of the nucleus, PHB2 can directly bind to DNA and RNA or interact with transcription factors, such as ERα, MyoD, MEF2, and HES1. It is reported that PHB2 can function as a transcription cofactor because it binds *PIG3* at the (TGYCC)_15_ motif. EMSA and gel shift assay confirmed that both PHB1 and PHB2 directly bind to *PIG3* promoter DNA. However, further studies are needed to confirm whether PHB1 and PHB2 directly regulate *PIG3* transcription.

### 6.4. PHB2 Modulators Block Translation Initiation

In a 2022 cell paper, it was reported that PHB2 is a common essential gene, and the main function is involved in protein translation [10]. There are two publications that support this observation. Two PHB2 specific modulators, rocaglamide [72] and fluorizoline [41,42], bind to PHB2 and inhibit protein translation. (1) Rocaglamide inhibits the Ras-Raf-MEK-ERK signaling pathway, suppressing Mnk-1-dependent phosphorylation of the translation initiation factor, eIF4E, which plays a crucial role in cap-dependent protein translation [85,86]. (2) Fluorizoline effectively inhibits protein synthesis by two mechanisms. One is by inhibiting the activity of key translation factors initiation factor 2 (eIF2) and elongation factor 2 (eEF2) due to a calcium influx alteration. The second is disrupting signals through mTOR1, a translational regulator complex. Specifically, fluorizoline prevents phosphorylation of 4E-BP1 and this hypophosphorylated 4E-BP1 binds and inhibits eIF4E. It was reported that eIF4E levels in cancer cells are much higher than in normal development, therefore, partially inhibiting eIF4E may be effective in killing cancer cells without being toxic to normal cells [87]. In addition, many members of the flavaglines family demonstrate protection of nonmalignant cells from chemotherapy-derived cytotoxicity [69]. Therefore, targeting PHB2 appears to be a safer and more sufficient cancer treatment strategy.

### 6.5. Rational Design of Targeting PHB2

PHB2 modulators are useful to understand PHB’s function in a variety of physiological and pathological conditions, including cancer. There are twelve reported PHB2 modulators, but most of them have only been tested in an in vitro system and are less potent [69]. Rocaglamide A has been tested in in vivo models of melanoma and lung cancer, illustrating a promising drug efficacy [73,74]. The increasing reports on PHB’s functions will encourage development of new PHB modulators and provide future clinical treatment strategies. Therefore, the rational design of PHB2 modulators is urgently needed to lead to breakthroughs in cancer therapy.

## 7. Conclusions

PHB1 and PHB2, belonging to the SPFH family, have been studied for over 30 years and have proved to be critical proteins with varying functions in different cellular locations, including the mitochondria, cell membrane, and nucleus. PHB2 is critical for organ development as loss of PHB2 leads to organ dysfunction. However, in many different cancer types, PHB2 is overexpressed and functions as either a tumor suppressor or an oncogene. As PHB2 is an oncogene in most cancer types, it is a promising target for cancer treatment. Although twelve PHB2 modulators have been developed, none of these modulators have been tested in clinical trials at this point. This leaves room for future research and development to inhibit PHB2’s critical function in cancer.

## Figures and Tables

**Figure 1 cells-12-01211-f001:**
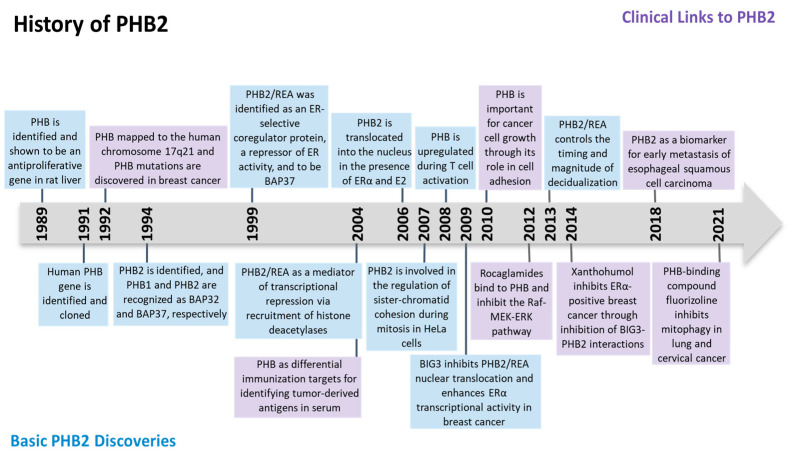
Timeline of major discoveries in prohibitin 2 (PHB2) studies, including basic science and clinical translational studies.

**Figure 2 cells-12-01211-f002:**
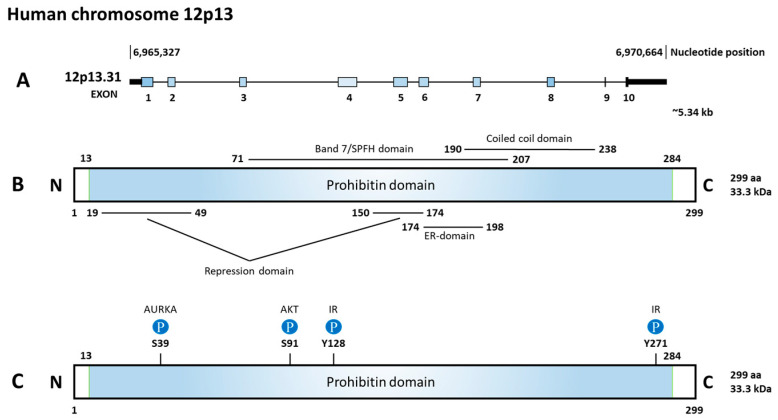
Diagram of gene, function domains, and phosphorylation sites of prohibitin 2 (PHB2). (**A**) *PHB2* is located on chromosome 12p13.31, consisting of 10 exons. (**B**) The prohibitin domain locates from 13aa to 284aa. Inside the prohibitin domain, there are two repression domains, a Band7/SPFH domain, a coiled coil domain, and an estrogen receptor (ER)-domain. (**C**) Upstream regulators and the phosphorylated site on PHB2. Aurora kinase A (AURKA) phosphorylates S39, AKT serine/threonine kinase (AKT) phosphorylates S91, and insulin receptor kinase (IR) phosphorylates Y128 and Y271.

**Figure 3 cells-12-01211-f003:**
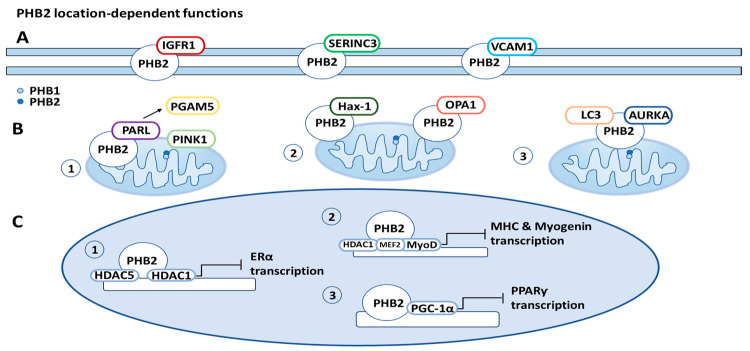
Prohibitin 2 (PHB2) location-related functions. (**A**) Cell membrane: PHB2 is associated with a variety of proteins on the cell membrane, including insulin-like growth factor-1 receptor (IGFR1), serine incorporator 3 (SERINC3), and vascular cell adhesion protein 1 (VCAM1). (**B**) Mitochondria: (1) PHB2 initiates mitophagy through binding presenilin-associated rhomboid-like protein (PARL), releasing PGAM5 (PGAM family member 5, mitochondrial serine/threonine protein phosphatase), and consequently stabilizing PINK1 (PTEN induced putative kinase 1). (2) PHB2 increases apoptosis resistance through binding optic atrophy 1 protein (OPA1) and HCLS1-associated protein X-1 (Hax-1). (3) PHB2 also initiates mitophagy through forming a tripartite complex with LC3 (microtubule-associated protein light chain 3) and aurora kinase A (AURKA). (**C**) Nucleus: (1) PHB2 recruits histone deactylases (HDACs) to the nucleus, inhibiting transcription of nuclear receptors, such as estrogen receptor-alpha (ERα). (2) PHB2 also interacts with HDAC1 (histone deacetylase 1) and forms a complex with transcription factors MEF2 (myocyte enhancing factor 2A) and MyoD (myoblast determination protein 1) to inhibit MHC (myosin heavy chain protein) and myogenin transcription. (3) PHB2 interacts with PGC-1α (peroxisome proliferator-activated receptor-ƴ coactivator), inhibiting the transcription of peroxisome proliferator-activated receptor (PPARƴ).

**Figure 4 cells-12-01211-f004:**
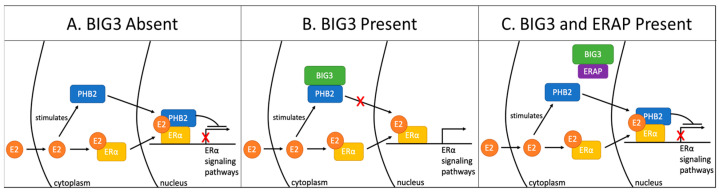
Interaction of PHB2, BIG3, and ERAP. (**A**) BIG3 Absent: E2 stimulates PHB2 to translocate into the nucleus and suppress the transcriptional activity of ER⍺. (**B**) BIG3 Present: BIG3 binds to PHB2 and inhibits its translocation into the nucleus, thereby preventing PHB2 inhibition of ER⍺ transcriptional activity. (**C**) ERAP Present: ERAP competitively binds BIG3, thus freeing PHB2, allowing its translocation into the nucleus.

**Figure 5 cells-12-01211-f005:**
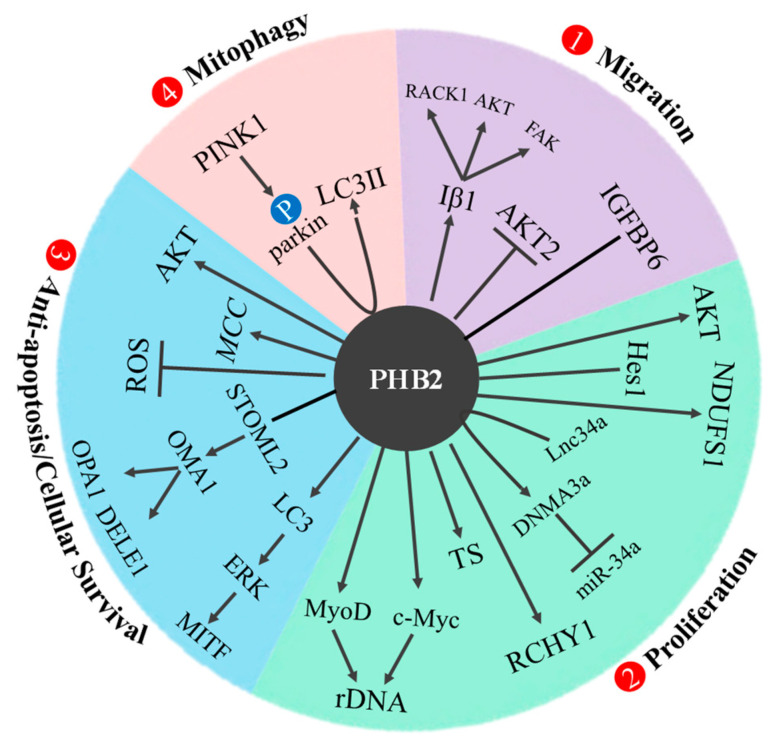
Schematic diagram of the multiple oncogenic mechanisms of PHB2. PHB2 has four main functions: (1) Promote migration, (2) Enhance proliferation, (3) Induce anti-apoptosis/cellular survival, and (4) Initiate mitophagy. The detailed PHB2 functions in different cancers were further described in Table 1.

**Table 1 cells-12-01211-t001:** (**A**) PHB2 tumor suppressor functions. (**B**) PHB2 oncogenic functions.

**(A)**		
	**Cancer Type**	**Mechanisms and References**
1	Breast cancer	Functions as a corepressor of ERα [34]. BIG3 overexpression represses PHB2’s translocation into the nucleus and its tumor suppressor function [36]. PGRMC1 prevents PHBs translocation into the nucleus and binding to Erα [38].
2	Osteosarcoma	BIG3 overexpression represses PHB2 [43].
3	Head and neck squamous cell	PHBs bind to the *PIG3* (P53-induced gene-3) promoter and initiate *PIG3* transcription and PIG3-mediated apoptosis [44].
**(B)**		
	**Cancer Type**	**Mechanisms and References**
1	Prostate	Inhibits AKT2 expression and promotes migration [45,46].
2	Non-small cell lung	Activates RACK1, AKT and FAK through integrin βI to promote migration [46]. Under mitochondrial stress, PINK1 phosphorylates parkin, causing the breakdown of OMM and exposure of IMM, leading to PHB2-LC3II mediated mitophagy [47].
3	Colon	Stabilizes thymidylate synthase (TS), promoting TS transcription. Rabdosianone I binds ANT2 and PHB2, causing TS degradation and inhibition of TS transcription [48]. PHB2 induces RCHY1 expression. DHA inhibits PHB2 and RCHY1 expression and suppresses p53 and p21 [49]. PHB2 binds NDUFS1 to stabilize mitochondria complex 1 and enhance its activity. This binding boosts OXPHOS levels and promotes cell proliferation and tumorigenesis [50]. Lnc 34a binds to the miR-34a promoter by recruiting PHB2/DNMA3a and HDAC1 to methylate miR-34a [51].
4	Hepatocellular	Lnc34a binds to PHB2, forming a complex with DNMT3A to methylate miR-34a. HDAC is consequently induced to inhibit miR-34a transcription [52]. A potential anticancer drug, PINT87aa, blocks PHB2 transcription, and prevents miR-34 methylation [53].
5	Leukemia	The PHB2-ANT2 complex stabilizes mitochondria. Loss of PHB2 induces apoptosis [54]. Fluorizoline (a PHB2 modulator), increases NOXA in the presence of PHB2, inducing p53-mediated apoptosis [55]. AKT1 and AKT2 phosphorylate PHB2 at S91 and regulate nuclear-mitochondria activity, further promoting cell survival [56].
6	Ovarian	Destruction of the PHB2/STOML2 complex promotes OMA1 protease activity, causing OPA1 and DELE1 cleavage, which contributes to apoptosis resistance [57].
7	Rhabdomyosarcoma	PHB2 promotes rDNA transcription through Myc-dependent rDNA transcription [20]. PHB2 decreases RMS differentiation through repressing MyoD via rDNA [20]. The PHB2-IGFBP6 complex enhances PHB2 phosphorylation at Y114 and Y259 which correlate with cell migration [58].
8	Multiple myeloma	MCC interacts with PHB2 at various levels, including activating c-Raf and the ERK1/2 pathway [59].
9	Esophageal squamous cell	Promotes cell proliferation by activating AKT, MMP9 and RAC1 [60].
10	Hematologic: lymphoid and myeloid	Promotes mitochondrial stabilization and prevents ROS-mediated cell death in hematologic malignancies [61].
11	Melanoma	Regulates LC3/ERK/MITF melanogenic signaling [62]. Mel9 and Mel41 inhibit AKT phosphorylation and downregulate PHBs to promote apoptosis [62].
12	Pancreatic	JI051 interacts with PHB2 and prevents Hes1’s translocation to the nucleus [63].

**Table 2 cells-12-01211-t002:** The reported modulators for PHB2.

	Ligand Name	Description	Target	Kd Potency	System	Effect of Binding	Mechanisms
1	Fluorizoline (FLZ)	A cytotoxic trifluorothiazoline	PHBs	1 µM	Lymphocytic leukemia cells	Inhibits epidermal growth factor (EGF)/RAS-induced C-RAF activation	A PHB1/2 specific modulator [41,42] Induces mitochondrial fragmentation, cristae disorganization and the intrinsic pathway of apoptosis in cancer cells [70]
2	Rocaglamide (RocA)	Flavagline	PHBs	50 nM	Breast cancer cells Mitochondria of cancer cells Osteoblasts Melanoma cells (human xenograft model). Lung cancer cells and xenografts	Inhibits the formation of pseudopodia, which are enriched in PHBs used by cancer cells to migrate [71] Inhibit mitophagy and energy production Promotes osteoblast differentiation Reduces MEK1/2 and ERK1/2 signaling, inhibiting melanoma cell growth, and inducing apoptosis Inhibit cell proliferation, migration, and tumor growth	Directly target PHB1 and PHB2 [72] Prevents metastasis of cancer cells [71] Inhibit the interactions between PARL and PHBs, the downregulation of PARL, PINK1 accumulation in depolarized mitochondria and the mitochondrial recruitment of Parkin Disrupts the interaction between PHB and CRAF in melanoma cells [73] Block EGF/RAS-mediated CRAF activation [74]
3	FL3	Flavagline	PHBs	50 nM	Mitochondria of cancer cells. Human cervical cancer cells and colon cancer cells	Inhibit mitophagy and energy production. Block cancer cell proliferation and energy production	Inhibits the interactions between PARL and PHBs, downregulate PARL, PINK1 accumulation, and the mitochondrial recruitment of Parkin [72,75] Inhibits PHB2-mediated mitophagy [25]
4	Mel9 and Mel41	Melanogenin analogs	PHBs	0.1 µM	Melanoma cancer cells	Triggers a cascade of events involving LC3, the kinase ERK and the transcription factor MITF	Mel9 and Mel41 binds PHB2 and inhibits p-ERK and p-AKT [62]
5	JI051	Synthesized Hes1 inhibiting compound	PHB2	EC_50_ of 0.3 μm	Pancreatic cancer	G2/M cell cycle arrest is promoted	JI051 binds to PHB2 to stabilize the interaction between Hes1 and PHB2 outside the nucleus [63]
6	Aurilide	Cyclic depsipeptide isolated from a marine mollusk	PHBs	3–6 nM	HeLa cells	Disintegration of mitochondria and apoptosis, leading to an extreme toxicity	Blocks the PHB-dependent inhibition of the proteolytic processing of the dynamin-like GTPase optic atrophy 1 (OPA1) [76]
7	Vi capsular polysaccharide (Vi)	Of *Salmonella typhi*, causative agent of typhoid fever	PHBs	1–5 µg/mL	Human intestinal epithelial cells	Blocks IL-2 secretion from T cells stimulated through the T-cell receptor (TCR) but not Protein Kinase C	Interacts with cell surface PHBs to inhibit ERK activity and interleukin-8 secretion in human intestinal epithelial cells [77]
8	Lipoteichoic acid	Polyanionic bacterial lipid, part of gram-positive bacteria cell walls	PHBs	Unknown	Human corneal keratocytes	Highlights the implication of PHBs in the immune response to infections	Lipoteichoic acid has been shown to activate RAF in the cornea, but whether this involves PHBs has not been documented yet [78]
9	Capsaicin	A component of hot chili peppers	PHB2	Too high to be pertinent for treatment	Leukemia	Initiates mitochondrial apoptosis in human myeloid leukemia cells	Promotes PHB2’s translocation from mitochondria to the nucleus, dissociating it from the Adenine Nucleotide Translocator 2 (ANT2) [79]
10	Xanthohumol	Polyphenol chalcone found in hops	PHB2	50 µM	Breast cancer cells	Block ERα-positive breast cancer cell growth in vitro and in vivo	Binds to PHB2 inhibiting its interaction with BIG3, which suppresses the inhibitory activity PHB2 has on the ERα in breast cancer cells [80]
11	PDD005	Purine derivative drug	PHBs	1.29 ± 1.16 µM for PHB2	Nervous system	Inhibits neuro-inflammation, levels of PHBs increase in mice when treated with PDD005 and decreases levels of the cytokine IL-1β	Enhances expression of PHBs in the hippocampal dentate gyrus of aged mouse and increases the phosphorylation of GSK-3β in organotypic hippocampal slice cultures, suggesting that this signaling protein is involved, at least partially, in the mechanism of neuroprotection [81,82]
12	DHA	Derivative of the antimalaria drug, Artemisinin	PHB2	20 µM	Colon cancer	Synergistic anticancer effects when with oxaliplatin by the promotion of PHB2 degradation in colon cancer cells	Downregulates PHB2 expression in a ubiquitylation-dependent manner, blocking PHB2-induced RCHY1 upregulation and the downregulation of p53 and p51 [49]

## Data Availability

Not applicable.

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
