# Peer review of "Essential Protein PHB2 and Its Regulatory Mechanisms in Cancer"

_cells, 2023, doi:10.3390/cells12081211_

Round 1

Reviewer 1 Report

I consider that this article fits well to the standard of Cells. The authors summarized well the importance of PHB2 as a therapeutic target against cancer. Considering the growing importance of prohibitins in cancers, this review provides very useful information. Nevertheless, few caveats needs to be corrected:

- it is better to replace ‘PHB2 inhibitors” by “PHB2 ligands” or “PHB modulators” since these compounds can inhibit one signaling pathway and activate another in the same time.

- the following sentence is misleading :”Fluorizoline effectively inhibits protein synthesis by two mechanisms. One is by inhibiting the activity of key translation factors via phosphorylation of 597 initiation factor 2 (eIF2) and elongation factor 2 (eEF2)”. Indeed, the inhibition of eIF2 and eEF2 is consecutive to a calcium influx, not to an interaction between PHB2 and these factors

- in figure 5, the text written in the cyan section should not be upside down.

Author Response

Reviewer 1

  1. I consider that this article fits well to the standard of Cells. The authors summarized well the importance of PHB2 as a therapeutic target against cancer. Considering the growing importance of prohibitins in cancers, this review provides very useful information. Nevertheless, few caveats needs to be corrected:

Response: We appreciate that the reviewer recognizes the value of our manuscript. Thank you for your comments, our response is followed point-to-point.

  1. it is better to replace “PHB2 inhibitors” by “PHB2 ligands” or “PHB modulators” since these compounds can inhibit one signaling pathway and activate another in the same time.

Response: Thank you for the suggestion. “PHB2 inhibitors” have been replaced by “PHB modulators”.

  1. the following sentence is misleading :”Fluorizoline effectively inhibits protein synthesis by two mechanisms. One is by inhibiting the activity of key translation factors via phosphorylation of 597 initiation factor 2 (eIF2) and elongation factor 2 (eEF2)”. Indeed, the inhibition of eIF2 and eEF2 is consecutive to a calcium influx, not to an interaction between PHB2 and these factors

Response: Thank you for your comments, we have taken your suggestion. Now on page 19, line 664-665, current version has been changed to “One is by inhibiting the activity of key translation factors initiation factor 2 (eIF2) and elongation factor 2 (eEF2) due to a calcium influx alteration.”

  1. in figure 5, the text written in the cyan section should not be upside down.

Response: Thank you for the suggestion. A new Figure 5 is provided that the text written in the cyan and green section is now flipped upright so that it is uniform across the sections.

Reviewer 2 Report

The article of Amanda Qi et al. describes the effect of PHB2 in cancer development. The mechanisms of action of PHB2 in different types of cancer are described.

The paper is well presented and well written.

However, there are minor comments on submitting the article.

1) No section conclusions. The authors should briefly generalize the information presented; it is possible to remake the discussion section. At the discretion of the authors.

2) either ko PHB2 or PHB2 ko ….. Check against the text and write in the same style

3) Check all abbreviations in the text, it is necessary that the decoding is indicated where it is mentioned for the first time

4) If the abbreviation is deciphered, it should not be entered again in another place in the text. Needs to be checked.

Author Response

Reviewer 2

The article of Amanda Qi et al. describes the effect of PHB2 in cancer development. The mechanisms of action of PHB2 in different types of cancer are described.

The paper is well presented and well written.

However, there are minor comments on submitting the article.

  1. No section conclusions. The authors should briefly generalize the information presented; it is possible to remake the discussion section. At the discretion of the authors.

Response: We appreciate the reviewer’s encouragement and the valuable suggestions. Now a new section, “6.6 Conclusion” was provided at the end of the manuscript to summarize couple of important points in this manuscript. Here is the section.

“6.6 Conclusion:

PHB1 and PHB2, belonging to the SPFH family, have been studied for over 30 years and have proved to be critical proteins with varying functions in different cellular locations including mitochondria, cell membrane and nucleus. PHB2 is critical for organ development as loss of PHB2 leads to organ dysfunction. However, in many different cancer types, PHB2 is overexpressed and functions as either a tumor suppressor or an oncogene. As an oncogene in most cancer types, PHB2 is a promising target for cancer treatment. Although twelve PHB2 modulators have been developed, none of these modulators have been tested in clinical trials at this point. This leaves room for future research and development to inhibit PHB2’s critical function in cancer.”

  1. either ko PHB2 or PHB2 ko ….. Check against the text and write in the same style

Response: Thank you for the suggestion. They have all been changed to “ko PHB2” to remain consistent.

  1. Check all abbreviations in the text, it is necessary that the decoding is indicated where it is mentioned for the first time

Response: Thank you for the suggestion. All abbreviations have been explained.

  1. If the abbreviation is deciphered, it should not be entered again in another place in the text. Needs to be checked.

Response: Thank you for the suggestion. We have corrected them.

Reviewer 3 Report

The manuscript entitled "Essential Protein PHB2 and Its Regulatory Mechanisms in Cancer" provides a comprehensive review of the structure of prohibitin 2 (PHB2), its site-dependent functions, its role in various cancers, its inhibitors, and its future.

Overall, the manuscript is well written and is easy to understand. The references used in the manuscript are recent and are adequate (34 of the 88 references are from the last five years). Regarding the novelty of the manuscript, as the authors state, there are many papers dealing with PHB1, while PHB2 is less explored. Although there are some reviews dealing with PHB2 and cancer, both are dated (2007 and 2015), so this review provides an updated and expanded insight into the mechanisms underlying PHB2 and its impact on the onset of different types of cancer,

In my opinion, the results shown in this review are interesting for a broader community.

Although it comes with some issues that need to be addressed:

Abbreviations: some abbreviations remain undefined in the manuscript V. gr. CCLE, tRNA, AAA proteases, SERINC3, IGFR1, VCAM1, HIV, PINK1-PRKN, MyoD and MEF2, PGC-1α, MCF7…, or are used before being defined. Though there is a list of abbreviations, it would be easier to understand if they were explained in the text. According to the instructions for authors:

Acronyms/Abbreviations/Initialisms should be defined the first time they appear in each of three sections: the abstract; the main text; the first figure or table. When defined for the first time, the acronym/abbreviation/initialism should be added in parentheses after the written-out form.

Once it is defined as knockout (ko) PHB2 in line 82, it is spelt as knockout PHB2, ko PHB2, and PHB2 k2 in the very next lines.

Figure 1 legend, use the full name of prohibitin 2.

Figure 2 legend, consider use the full name of PHB2, ER, AURKA, AKT and IR.

Figure 3 legend, consider use the full name of PHB2, IGFR1, SERINC3, VCAM1, PARL, PGAM5, PINK1, OPA1, Hax-1, LC3, AURKA, HDACs, Erα, HDAC1, MEF2, MyoD, MHC, PGC-1α and PPARƴ.

Table 1: the first two types of cancer mechanisms are not numbered. Ovarian cancer starts with number 4; numbers 8 to 10 are also not numbered as well, and the numbers in melanoma are in different font size.

Table 2: separate units from the numbers; In the effect of binding column some inhibitors are numbered, and others are not.

Revise the References section as it does not follow the instruction for authors. Remove the p from the references page range and use italics on the abbreviated journal name to match the instructions for authors. Revise references 38 and 51 as they appear to be missing the page range. It should look like this:

Journal Articles:

1. Author 1, A.B.; Author 2, C.D. Title of the article. Abbreviated Journal Name Year, Volume, page range.

Best regards

Author Response

Reviewer 3

The manuscript entitled "Essential Protein PHB2 and Its Regulatory Mechanisms in Cancer" provides a comprehensive review of the structure of prohibitin 2 (PHB2), its site-dependent functions, its role in various cancers, its inhibitors, and its future.

Overall, the manuscript is well written and is easy to understand. The references used in the manuscript are recent and are adequate (34 of the 88 references are from the last five years). Regarding the novelty of the manuscript, as the authors state, there are many papers dealing with PHB1, while PHB2 is less explored. Although there are some reviews dealing with PHB2 and cancer, both are dated (2007 and 2015), so this review provides an updated and expanded insight into the mechanisms underlying PHB2 and its impact on the onset of different types of cancer,

 In my opinion, the results shown in this review are interesting for a broader community.

 Although it comes with some issues that need to be addressed:

  1. Abbreviations: some abbreviations remain undefined in the manuscript V. gr. CCLE, tRNA, AAA proteases, SERINC3, IGFR1, VCAM1, HIV, PINK1-PRKN, MyoD and MEF2, PGC-1α, MCF7…, or are used before being defined. Though there is a list of abbreviations, it would be easier to understand if they were explained in the text. According to the instructions for authors:

Response: Thank you for your suggestions. All the listed abbreviations are now defined the first time they are introduced in the manuscript.

  1. Acronyms/Abbreviations/Initialisms should be defined the first time they appear in each of three sections: the abstract; the main text; the first figure or table. When defined for the first time, the acronym/abbreviation/initialism should be added in parentheses after the written-out form.

Response: Thank you for your suggestions. All the listed abbreviations are now defined the first  time they are introduced in each of three sections.

  1. Once it is defined as knockout (ko) PHB2 in line 82, it is spelt as knockout PHB2, ko PHB2, and PHB2 k2 in the very next lines.

Response: Thank you for the suggestion. It has been corrected so that after the spelling of knockout (ko) PHB2, “ko PHB2” is used from then on out.

  1. Figure 1 legend, use the full name of prohibitin 2.

Response: Thank you for the suggestion. The PHB2 is corrected to prohibitin 2 in the legend of Figure 1.

  1. Figure 2 legend, consider use the full name of PHB2, ER, AURKA, AKT and IR.

Response: Thank you for the suggestion. The abbreviations of PHB2, ER, AURKA, AKT and IR are changed to their full names.

  1. Figure 3 legend, consider use the full name of PHB2, IGFR1, SERINC3, VCAM1, PARL, PGAM5, PINK1, OPA1, Hax-1, LC3, AURKA, HDACs, Erα, HDAC1, MEF2, MyoD, MHC, PGC-1α and PPARƴ.

Response: Thank you for the suggestion. The abbreviations of PHB2, IGFR1, SERINC3, VCAM1, PARL, PGAM5, PINK1, OPA1, Hax-1, LC3, AURKA, HDACs, Erα, HDAC1, MEF2, MyoD, MHC, PGC-1α and PPARƴ  are changed to their full names.

  1. Table 1: the first two types of cancer mechanisms are not numbered. Ovarian cancer starts with number 4; numbers 8 to 10 are also not numbered as well, and the numbers in melanoma are in different font size.

Response: Thank you for the suggestion. The cancer mechanisms that were not numbered are now  all numbered correctly. We also correct the font size.

  1. Table 2: separate units from the numbers; In the effect of binding column some inhibitors are numbered, and others are not.

Response: Thank you for the suggestion. The inhibitors are now all numbered in both the effect of binding column and the mechanism column if there is more than one effect of binding or mechanism. The numbers have also been separated from the units. However, we hope the layout orientation for this table can be kept as original landscape, not the current portrait, to allow more words to fit in each line.

  1. Revise the References section as it does not follow the instruction for authors. Remove the p from the references page range and use italics on the abbreviated journal name to match the instructions for authors. Revise references 38 and 51 as they appear to be missing the page range. It should look like this:

Journal Articles:

  1. Author 1, A.B.; Author 2, C.D. Title of the article. Abbreviated Journal Name Year, Volume, page range.

Response: We appreciate the suggestions. Now the EndNote style of “MDPI ACS Journal” has been applied to all the references. We also have fixed the page number for references 38 and 51.